



# Assessment of the Paris urban heat island in ERA5 and offline SURFEX-TEB (v8.1) simulations using METEOSAT land surface temperature product

Miguel Nogueira[1], Alexandra Hurduc[1], Sofia Ermida[1,2], Daniela C. A. Lima[1], Pedro M. M. Soares[1], Frederico Johannsen[1], Emanuel Dutra[1,2]

[1]Instituto Dom Luiz, IDL, Faculty of Sciences, University of Lisbon, 1749-016 Lisbon, Portugal
[2]Instituto Português do Mar e da Atmosfera, IPMA, 1749-077 Lisbon, Portugal

*Correspondence to*: Frederico Johannsen (jfjohannsen@fc.ul.pt)

**Abstract.** Cities concentrate people, wealth, emissions, and infrastructures, thus representing a challenge and an opportunity for climate change mitigation and adaptation. This places an urgent demand for accurate urban climate projections to help organizations and individuals making climate smart-decisions. However, most of the state-of-the-art global and regional climate models have an oversimplified representation of (or completely neglect) urban climate processes. Here, we use the city of Paris as a case study to show that this is the case for the fifth (and latest) generation reanalysis from the European Centre for Medium-Range Weather Forecasts (ERA5) and for simulations employing the widely used bulk bare rock approach to urban climate parameterization. Subsequently, we leveraged on the hourly resolution of ERA5 and the Satellite Application Facility Land Surface Analysis (LSA-SAF) land surface temperature product to demonstrate the significant added value of employing the SURFEX land-surface model coupled to Town Energy Balance (TEB) urban canopy model in simulating the Parisian Surface Urban Heat Island (SUHI) during daytime and the urban heat island during both daytime and nighttime. Our results showed the significant added value of SURFEX-TEB in reproducing the observed daytime and nighttime Parisian urban heat island effect. An annual average bias magnitude reduction of 0.5°C was observed for daytime and around 1.5°C for nighttime when compared to ERA5 and bare rock approach. Also, SURFEX-TEB revealed an overall better performance in reproducing the observed daytime SUHI, whilst the added value of SURFEX-TEB was lower during nighttime (but still slightly better than ERA5 and the bare rock approach), due to the lack of land-atmosphere feedbacks in the proposed offline framework. Finally, the offline SURFEX-TEB framework applied here demonstrates the ability to simulate the urban climate, which is an asset to build urban climate projections that allow the development of mitigation and adaptation strategies.

## 1 Introduction

Urban areas accommodate nearly half of the global population, and this fraction is projected to increase to 68% by 2050 according to the World Health Organization (WHO, 2018). Moreover, cities concentrate wealth, infrastructures, and





emissions - being responsible for about 75% of the global greenhouse gas emissions from energy consumption (IPCC, 2014). Consequently, understanding and simulating the urban climate evolution is a key task for climate change assessments and for designing climate change adaptation and mitigation strategies.

The urban areas are characterized by drastic land-use changes which are responsible for increased trapping and absorption of solar radiation, reduced evapotranspiration, and decreased nighttime cooling in built-up areas. As a result, cities typically

have warmer air and surface temperatures compared to nearby rural environments. This is the well-known urban heat island (UHI) effect, which has been found over multiple cities across the globe (see e.g., Deilami et al., 2018 for a recent review).

The identification and quantification of the UHI dynamics have proven to be challenging. Despite its widespread emergence in urban environments, the UHI is sensitive to the specific land surface characteristics and meteorological conditions, hence displaying significant variability between different locations and periods. Indeed, previous investigations reported several

different relevant UHI dependencies, including city size and population density (Oke, 1982; Clinton & Gong, 2013; Manoli et al., 2019), urban vegetation coverage (Kaloustian & Diab, 2015; Peng et al., 2012; Zhou et al., 2014; Nogueira & Soares, 2019), background climate conditions (namely precipitation and wind, Zhou et al., 2013; Lemonsu et al., 2013; Zhao et al., 2014; Manoli et al., 2019) and urban morphology (e.g., city geometry, building height, construction materials, etc., Oke, 1973; 1982; Zhou et al., 2017; Krayenhoff et al., 2018; Nogueira & Soares, 2019; Masson et al., 2020). Heat release

resulting from human activities has also been shown to modulate the UHI (De Munck et al., 2013; Schoetter et al., 2020). Moreover, surface and near-surface air temperature over "natural" regions also display large sensitivity to the complex land use and land cover patterns (e.g, Beljaars et al., 1996; Koster et al., 2004; Johannsen et al., 2019; Nogueira et al., 2020a, 2021), which represents an additional layer of complexity to the UHI.

Investigations of the UHI based on contrasting in situ temperature and surface fluxes observations from cities and

neighboring rural locations are generally unable to capture its complex spatial heterogeneity, particularly for large cities, resulting in large uncertainties in the UHI characterization (Stewart, 2011; Schwarz et al., 2011; Stewart and Oke, 2012). The development of dense urban meteorological station networks allowed to partially overcome these limitations, but the temporal and spatial coverage of such networks remains too narrow to fully characterize the urban induced climate modulation (Muller et al., 2013; Konstantinov et al., 2018). Remote sensing techniques provide a widely used alternative for

comprehensive characterization of the UHI and its variability, providing reliable estimates for numerous land surface properties with wide spatial coverage and adequate spatial and temporal sampling, including land surface temperature (LST), land use and land cover (LULC) maps, soil moisture, rainfall, and snow, amongst others (see Balsamo et al., 2018 for a recent review).

Numerous works revealed the existence of a surface urban heat island (SUHI), referring to warmer LST in urban areas

compared to its rural environment (e.g., Roth et al., 1989; Imhoff et al., 2010; Schwarz et al., 2011; Peng et al., 2012; Zhao et al., 2014; Zhou et al., 2017). Yet, these studies identified significant differences between the UHI and SUHI, including the maximum UHI hour and seasonality, and the relationship between thermal contrast magnitude and land use. Moreover, LST estimates are often restricted to clear-sky conditions since, typically, the available all-sky estimates are restricted by very





coarse spatial resolution which is inappropriate to characterize the urban environments (Masson et al., 2020). The LST
estimates are also often constrained by the time of satellite overpass, which limits the temporal resolution.

Urban climate simulations generated by physically-based numerical models can potentially circumvent some of the
limitations of in situ and remote sensing observational products. Specifically, coherent information for multiple relevant
variables with high spatial and temporal coverage and resolutions may be obtained. Additionally, due to the complexity and
diversity of cities around the world, the city scale climate properties are specific and often limited to a particular location.
Moreover, while observations cover the past, numerical simulations can be extended to the future and, therefore, consider
different scenarios of future socio-economic evolution, urban development, and adaptation strategies.

Most state-of-the-art climate models do not consider or have simplified representations of the urban environments (Garuma,
2018; Zhao et al., 2021). Furthermore, the available large ensembles of Earth System Models (ESMs) and Global Climate
Models (GCMs) typically have coarse spatial resolutions (~100 km), which are inadequate for representing most of the city-
scale processes. Typically, state-of-the-art large multi-model ensembles of Regional Climate Models (RCMs) have grid-
resolutions on the order of tens of kilometers which is still inappropriate to simulate many aspects of the urban climate
system (e.g., Langendijk et al., 2019; Nogueira et al., 2020b; McNorton et al., 2021). The next generation of RCM
ensembles will have a resolution of a few kilometers, allowing a better simulation of the local climate variability (Jacob et
al., 2020). Indeed, several pilot studies have suggested significant added value in including urban canopy models (UCMs) to
parameterize interactions between the urban surface and the atmosphere in RCMs with resolutions of a few kilometers (Chen
et al., 2011; Kusaka et al., 2012; Hamdi et al., 2012; Lemonsu et al., 2014; Daniel et al., 2019; Garuma, 2018; Schoetter et
al., 2020). However, the use of UCM coupled to RCMs is not a standard procedure for climate simulation (and is not
projected to be in the next generation of multi-model RCM ensembles) due to its very high computational costs, resulting in
a poor representation of many aspects of urban climate in state-of-the-art RCM ensemble datasets (Langendijk et al., 2019;
Nogueira et al., 2020b).

The use of land-surface models (LSM) coupled to a UCM, forced offline by atmospheric data, provides a computationally
efficient option for urban climate simulation. This approach overcomes the computational and resolution limitations of
ESMs, GCMs and RCMs, but comes at the cost of neglecting the urban land-atmosphere feedbacks, providing only
diagnostics for the surface and near-surface variables. One may estimate the urban impact on surface and near-surface air
temperature and humidity, near-surface wind, latent and sensible heat fluxes, but not on clouds, precipitation, or local
circulations. Despite those limitations, recent studies have demonstrated the added value of this approach in reproducing key
features of observed urban climate compared to traditional climate simulations (without representation of urban processes),
including the UHI and the frequency, intensity and duration of urban extreme temperature events (Broadbent et al., 2018;
Conlon et al., 2016; Daniel et al., 2018; Kaloustian and Diab, 2015; Lemonsu et al., 2013, 2015; Nogueira and Soares, 2019;
Hamdi et al., 2020; Viguié et al., 2020; Nogueira et al., 2020b). Leveraging the competitive computational cost of offline
LSM-UCM simulations, these studies explored the local climate response to multiple different urbanization patterns and
emission scenarios over relatively long periods and at high spatial resolution. Additionally, Nogueira and Soares (2019)





demonstrated how this type of framework may be used to disentangle the impact of land-use change, from large-scale warming induced by greenhouse gas emissions, and from natural climate variability. This represents a critical task for anthropogenic climate change attribution and for designing effective mitigation strategies. The added value of the offline framework has also been demonstrated in simulating the impact of changes in vegetation cover patterns over non-urban regions (Johannsen et al., 2019; Nogueira et al., 2020a, 2021).

The present study assesses the ability of the LSM-UCM approach to downscale ERA5 reanalysis, the fifth, and latest, generation reanalysis from the European Centre for Medium-Range Weather Forecasts to resolutions of a few kilometers over dense urban areas. Specifically, we analyze the added value of the Météo-France SURFEX (Surface Externalisée) surface modelling platform (Le Moigne, 2018) in improving the simulation of the UHI and SUHI over Paris, a European mega-city characterized by a well-known strong urban heat island effect (e.g., Sarkar & De Ridder, 2011; De Munck et al., 2013; Hamdi et al., 2015; Lemonsu et al., 2015; Daniel et al., 2019). SURFEX is particularly relevant in this context since it has shown to perform particularly well in offline urban simulations (e.g., Hamdi et al., 2015; Lemonsu et al., 2015; Nogueira & Soares, 2019; Nogueira et al., 2020b). Previously, Nogueira et al. (2020b) used the offline LSM approach to perform downscaling of the EURO-CORDEX simulation ensemble for the historical and future periods over an urban grid-point inside the city of Lisbon and a neighboring rural grid-point. The results highlighted the poor representation of the UHI effect in the EURO-CORDEX RCMs and suggested the added value of the online approach for simulating the UHI effect. However, this study was limited to two single-column simulations and focused on the daily maximum and daily minimum 2-meter air temperature, which are diagnostic variables strongly constrained by the temperature forcing in the offline LSM approach that does not allow the atmospheric response induced by the changes in the fluxes. Here, we use high-resolution LST satellite data to investigate the spatial structure of the Parisian SUHI and its diurnal cycle in ERA5 and to assess the added value of the offline LSM downscaling approach, contrasting against the often-used bulk bare rock urban parameterization approach.

## 2. Methods

### 2.1 Observations and Reanalysis

Parisian daily maximum and minimum temperatures (respectively $T_{max}$ and $T_{min}$) were obtained from two weather stations (Fig. 1), retrieved from the Global Summary of the Day (GSOD), produced by the National Climatic Data Center (NCDC), which includes quality control checks and random error removal. The first station located in the Montsouris public park (48.82N, 2.33E), in Paris city center, was used as reference to characterize the Parisian urban temperature. The second station located in Melun (southeast of Paris - 48.61N, 2.67E), in a natural environment, was used as reference to characterize the Parisian surroundings temperature. These two stations were also previously employed by Hamdi et al. (2015) and Daniel et al. (2019) to characterize the Parisian UHI.

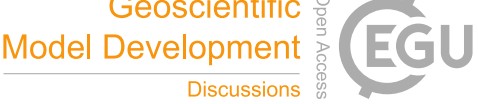

The LSA-SAF LST estimates are derived from the outgoing thermal infrared radiation (TIR) measured at top-of-atmosphere by the Spinning Enhanced Visible and InfraRed Imager (SEVIRI) onboard the Meteosat Second Generation (MSG) series by employing a generalized "split-window" technique (Freitas et al., 2010). The TIR spectral band (8–13 µm) is particularly appropriate as it presents relatively weak atmospheric attenuation under clear-sky conditions and includes the peak of the Earth's spectral radiance (Li et al., 2013; Ermida et al., 2019). The LSA-SAF LST estimates were available every 15 minutes from 2004 to present-day over land pixels within the MSG disk, comprising satellite zenith view angles between 0º and 80º,

with a 3 km resolution at the nadir. The LSA-SAF LST estimates were aggregated as the average at 00, 15, 30, and 45 minutes for each hour. Then, the hourly mean for the period ranging from 2004 through 2018 was computed. LST obtained through remote sensing is intrinsically directional due to the heterogeneity of the land surface. Still, given the model resolution considered in this study, LST's ability to evaluate model data should not be affected by its directional property.

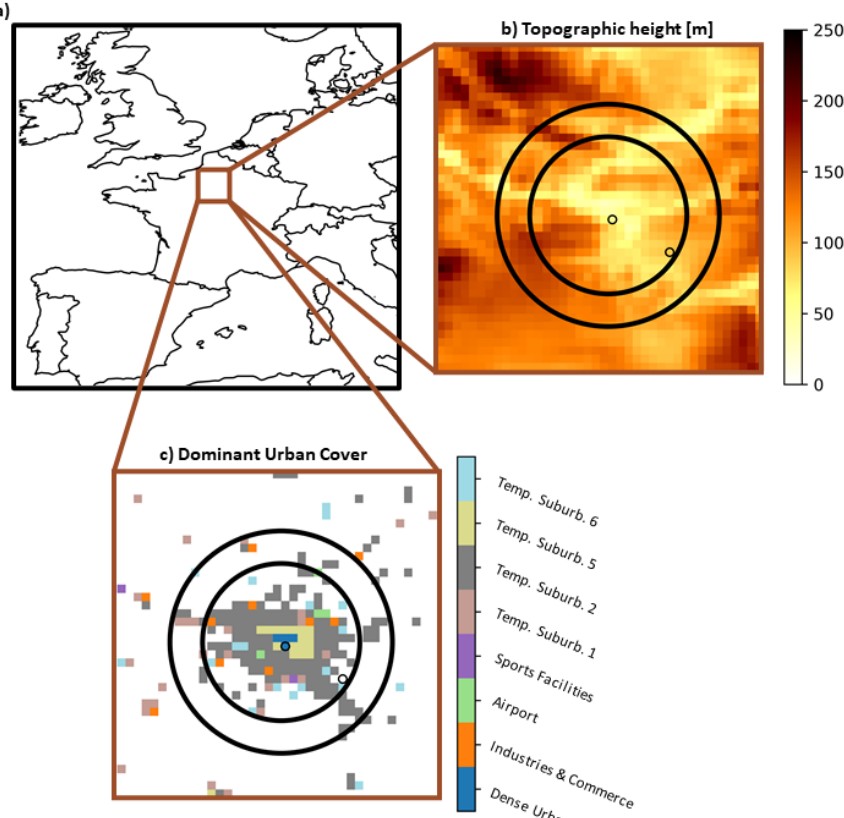

**Figure 1: Study domain identified by the brown square in panel a). Panel b) shows a zoom on the simulation domain with the color shading representing topographic height. Panel c) shows a zoom on the simulation domain with the color scheme representing the dominant urban classes for grid boxes where urban fraction exceeds 0.1 (grid boxes where urban fraction is below 0.1 are painted white). The large black circles in b) and c) identify the inner and outer rings for computing the SUHI (see Section 2.3). The 'o' markers in b) and c) identify the station locations for computing the UHI (see Section 2.3).**



ERA5 is the last-generation global atmospheric reanalysis produced by the ECMWF, extending from 1979 to the present (although a preliminary version of an extension to 1950 is already available). ERA5 is based on a recent version of the ECMWF Integrated Forecast System (IFS cycle 41r2), including several improvements compared to the version used in ERA-Interim (the ECMWF's previous generation reanalysis, Dee et al., 2011). Namely, ERA5 features increased temporal, horizontal and vertical resolutions (respectively 1 hour, ~31 km and 137 vertical levels extending from surface to 0.01 hPa) (see Hersbach et al., 2020 for a detailed description of ERA5), and an increased number and more recent versions of a wide variety of observational datasets are assimilated. Additionally, ERA5 benefits from improvements to several model parameterizations (e.g. convection and microphysics) and to the four-dimensional variational data assimilation scheme. Furthermore, it also presents an overall better accuracy in representing several climate variables compared to ERA-Interim, including LST, near-surface air temperature, wind, radiation, and rainfall (e.g., Urraca, 2018; Beck et al., 2019; Johannsen et al., 2019; Rivas and Stoffelen, 2019; Nogueira, 2020). ERA5 does not represent urban areas. McNorton et al (2021) proposed a new urban scheme for the ECMWF model showing that only large cities had some signal at 9km resolution, with a negligible impact at 30km resolution, which is similar to ERA5. However, ERA5 assimilates observations from ground weather stations (including 2-meter temperature, snow depth and relative humidity), and remote sensing data (Hersbach et al., 2020). Therefore, ERA5 should still be able to represent some sort of urban signal over large city-gridpoints.

## 2.2 SURFEX simulations

We performed a set of high-resolution (0.05°×0.05°) simulations of Paris and its surroundings (cf. Fig. 1) using the Météo-France SURFEX (Surface Externalisée) version 8.1 (Le Moigne, 2018) modelling platform. SURFEX couples multiple physical-based models over all types of natural surfaces, including the Interaction between Soil Biosphere and Atmosphere (ISBA) land-surface scheme over natural land surfaces (Calvet et al., 1998; Gibelin et al., 2006) and the Town Energy Balance (TEB) Urban Canopy Model (UCM) over urban surfaces. TEB uses the urban canyon approach (Oke, 1987) to simulate key urban physical processes on the local climate, including the possibility to account for the effects of vegetation and water bodies (see Masson et al., 2000 and Masson et al., 2013 for a detailed description of TEB). The SURFEX simulations were performed in an offline setup forced by ERA5 fields – namely surface pressure, precipitation, short- and long-wave radiative fluxes, and air temperature, humidity, and wind speed at 40 m height above sea level (above the Parisian urban canopy height). The simulations started in January 2003 and extended until the end of 2018, with a 15-minute time-step.

Two different SURFEX experiments were carried here. In the first one, denoted SFX-ROCK, the city grid boxes were described as rock covers. This bulk urban parameterization is often employed in state-of-the-art regional climate simulations (e.g., Daniel et al., 2019; Langendijk et al., 2019; Davin et al. 2019; Nogueira et al., 2020b). The second experiment, denoted SFX-TEB, employed the TEB UCM for urban grid boxes. Both experiments considered a multilayer soil diffusion scheme with 14 soil layers and a single-level canopy layer, which has previously been demonstrated to be adequate over European mid-rise cities such as Paris (Schoetter et al., 2020).





### 2.3 Assessment of the simulated UHI and SUHI

The UHI was defined here as the 2-meter air temperature ($T_{2m}$) difference between the urban (Montsouris) and the rural
(Melun) station locations. For ERA5 and SURFEX simulations this was estimated using the respective nearest-neighbor grid
boxes. Although the two-point difference approach cannot account for the complex spatial heterogeneity of urban
environments and their surroundings, the limited number of observations available to this study defined this particular
choice. Nonetheless, the complex spatial heterogeneity of the Paris area was accounted by the SUHI definition considered
here. Following the methodology employed in previous works (Peng et al., 2012; and Zhou et al., 2013, 2017) the SUHI was
defined as the difference between the average temperature within the considered urban cluster and the average temperature
within an equal area belt around it. This approach combined land cover data (from ECOCLIMAP-II) with LSA-SAF LST.
The urban cluster was defined as the grid boxes with urban fraction greater than 66% within the inner circle shown in Fig. 1.
The surrounding belt is also shown in Fig. 1. The SUHI was computed for ERA5, SFX-ROCK and SFX-TEB using the same
approach, considering the same urban fractions from the ECOCLIMAP-II land cover data for all datasets.
The daytime and nighttime UHI and SUHI were evaluated separately. The daytime maximum $T_{2m}$ and LST (denoted $T_{max}$
and $LST_{max}$ respectively) were computed as the maximum temperature within the 11 to 18 UTC interval. The nighttime
minimum $T_{2m}$ and LST (denoted $T_{min}$ and $LST_{min}$ respectively) were computed as the minimum temperature over the 00 to
07 UTC interval. We computed two error metrics for evaluating the UHI and SUHI in the different model-based datasets.
The first was the mean bias calculated following Eq. (1):

$$Bias = \frac{1}{N}\sum_{i=1}^{N}(m_k - o_k),\tag{1}$$

where $m_k$ and $o_k$ are respectively model simulated and observed values and N is the total number of days in the historical
time-series. The mean bias measures the models' systematic errors. The second was the Perkins skill score (Perkins et al.,
2007), henceforth denoted S, which measures the models' ability to reproduce the observed probability distribution functions
(PDFs):

$$S = 100 \times \sum_{i=1}^{B} \min[Z_{m,i}, Z_{o,i}],\tag{2}$$

where $\min[x, y]$ represents the minimum between two values, $Z_m$ and $Z_o$ are the modeled and observed empirical PDFs,
respectively, and B is the total number of bins used to compute the empirical PDFs (here we used steps of 1ºC). S provides a
measure of similarity between modeled and observed empirical PDFs, with S=100% if the model reproduces the empirical
PDF perfectly and decreasing towards zero as the similarity between the PDFs decreases. Both error metrics were also
applied to evaluate the daytime and nighttime LST over the study domain grid boxes and the daytime and nighttime $T_{2m}$ at
the two station locations.





## 3. Results

### 3.1 Intercomparison of the simulated SUHI over Paris

The observed LST averaged over the 2004-2018 period showed a clear signature of the Parisian SUHI effect during daytime
(Fig. 2a) and nighttime (Fig. 2b). The SUHI was not reproduced by ERA5 during daytime (Fig. 2c) nor nighttime (Fig. 2d).
The results also highlighted that ERA5 0.25º resolution is too coarse to reproduce the complex urban climate patterns, even
for a relatively large city as Paris. The simulation SFX-ROCK also failed to reproduce the Paris SUHI during daytime (Fig.
2e) and nighttime (Fig. 2f). As expected, the higher resolution did improve the simulation of topographic effects on surface
temperature south and northwestern of Paris (cf. Fig. 1b), although, the bare rock approach misrepresented the city LST
modulation. In contrast, the LST patterns resulting from the SFX-TEB simulation showed a clear signature of the Paris SUHI
during daytime (Fig. 2g), which was closer to the observed pattern. During nighttime, SFX-TEB shows some signature of the
SUHI (Fig. 2h) but underestimates its magnitude.

The improved ability of SFX-TEB in reproducing the urban LST over Paris was evidenced by the large reduction of the
median |Bias| over urban grid boxes (i.e., where the urban fraction was above 2/3), from 7.0ºC in ERA5 and 6.7ºC in SFX-
ROCK to 2.8ºC in SFX-TEB (Fig. 3a). Over natural surfaces, this reduction was lower, from 2.5ºC in ERA5 to 1.5ºC in
SFX-ROCK and 1.3ºC in SFX-TEB. The 0.2ºC difference between SFX-ROCK and SFX-TEB was due to the considered
definition of natural surfaces, encompassing all grid-boxes with urban fractions below 1/3. Indeed, in cases where the urban
fraction was zero, the SFX-ROCK and SFX-TEB simulations were identical. Notice, however, that the differences amongst
different datasets over natural surfaces were within the typical uncertainty associated with LSA-SAF LST estimates, which
is of the order of 2ºC (Trigo et al., 2015). Over mixed surfaces (i.e., urban fractions between 1/3 and 2/3) the median |Bias|
for annual averaged LST was 3.8ºC for ERA5, 3.3ºC for SFX-ROCK and 1.7ºC for SFX-TEB. These differences were also
within the typical observational uncertainty. Yet, the large improvements over urban surfaces and identical performances
over natural surfaces suggest that SFX-TEB represented an improvement over mixed surfaces too, particularly when the
grid-box urban fraction approaches 2/3.

The largest reductions to daytime LST over urban areas' systematic errors occurred during MAM, where the median |Bias|
was 8.8ºC for ERA5, 7.6ºC for SFX-ROCK, and 2.1ºC for SFX-TEB (Fig. 3e). A large reduction of the daytime LST
systematic error was also found during SON, where the median |Bias| was 6.5ºC for ERA5 and SFX-ROCK, and 0.8ºC for
SFX-TEB (Fig. 3i). During JJA, the median |Bias| was 8.2ºC for ERA5, 5.2ºC for SFX-ROCK, and 5.1ºC for SFX-TEB (Fig.
3g). Finally, during DJF the median |Bias| was 4.1ºC for ERA5, 6.3ºC for SFX-ROC, and 2.8ºC for SFX-TEB (Fig. 3c).
Notice that, on the seasonal scale, SFX-TEB was not always the best performing model during daytime (for example, ERA5
was the best performing model during DJF over natural surfaces, and SFX-ROCK was the best performing during JJA over
mixed surfaces).

During nighttime, the differences in annual average LST systematic errors amongst different simulations (Fig. 3b) were
lower than during daytime. Specifically, over urban surfaces, the median |Bias| was 1.8ºC in ERA5, 2.0ºC in SFX-ROCK,



and 1.7ºC in SFX-TEB. Over mixed surfaces, the median |Bias| was 0.7ºC in ERA5 and 0.6ºC in SFX-ROCK and SFX-TEB. Finally, over natural surfaces, the nighttime median |Bias| was 1.0ºC in ERA5 and 0.7ºC in SFX-ROCK and SFX-TEB. Notice that these differences amongst simulations were within the observational uncertainty. This was also true for all systematic differences in nighttime LST amongst different datasets on the seasonal scale (Fig. 3d, f, h, j).

Figure 4 evidences the clear reduction of the simulated LST systematic errors during daytime hours in SFX-TEB compared

to ERA5 and SFX-ROCK over urban (Fig. 4c) and mixed (Fig. 4b) grid boxes. Over natural surfaces, the daytime performance was similar for SFX-ROCK and SFX-TEB, both slightly outperforming ERA5 (Fig. 4a). Finally, during the night hours, Figure 4 showed a similar performance in reproducing the average LST over all surface types.





**Figure 2: Maps of LST averaged over the 2004-2018 period during daytime (left column) and nighttime (right column) over the study domain computed from LSA-SAF (a) and b)), ERA5 (c) and d)), SFX-ROCK (e) and f)), and SFX-TEB (g) and h)).**



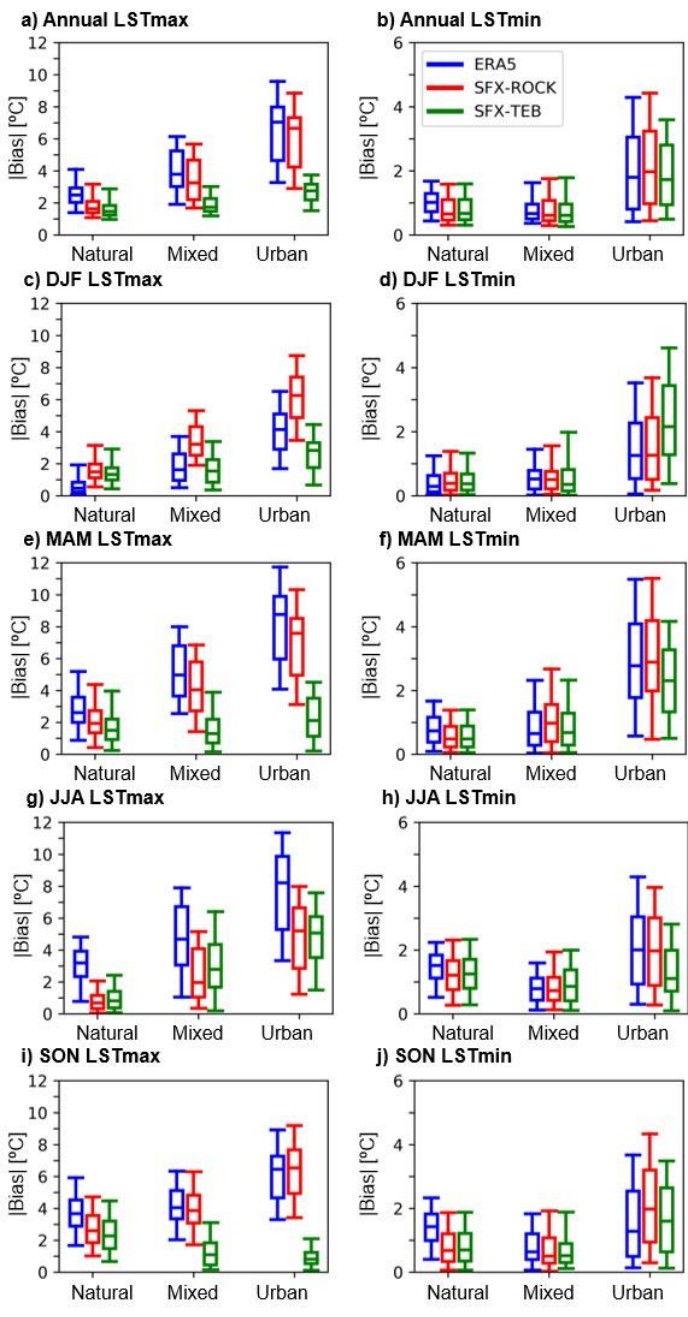

**Figure 3: Boxplots of absolute bias computed over the 2004-2018 period for daytime (left column) and nighttime (right column) for ERA5 (blue), SFX-ROCK (red), and SFX-TEB (green). The boxplots represent the bias spread for grid boxes classified as natural surfaces (grid boxes with urban fraction below 0.33), mixed surfaces (grid boxes with urban fraction between 0.33 and 0.66), and urban surfaces (grid boxes with urban fractions above 0.66). From top to bottom, the rows represent the bias computed for the full annual cycle, DJF, MAM, JJA and SON.**




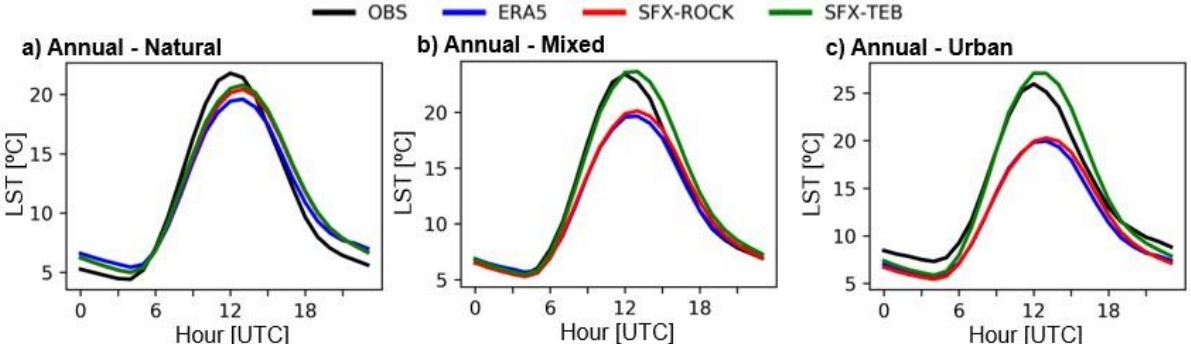

**Figure 4: Average diurnal cycle of LST computed over the 2004-2018 period from LSA-SAF (black), ERA5 (blue), SFX-ROCK (red), and SFX-TEB (green) for grid boxes classified as a) natural surfaces (grid boxes with urban fraction below 0.33), b) mixed surfaces (grid boxes with urban fraction between 0.33 and 0.66), and c) urban surfaces (grid boxes with urban fractions above 0.66).**

The SFX-TEB ability to reproduce the observed annual daytime LST PDF was better than ERA5 and SFX-ROCK over all surface types (Fig. 5a). Over urban surfaces, the median S score for daily maximum was 42% in ERA5, 49% in SFX-ROCK, and 62% in SFX-TEB. Over mixed surfaces, the median S score for daily maximum LST was 58% in ERA5, 64% in SFX-ROCK, and 67% in SFX-TEB. Over natural surfaces, the median S score for daily maximum LST was 66% in ERA5, 69% in SFX-ROCK, and 70% in SFX-TEB. The largest improvements in daily maximum LST S score associated with SFX-TEB emerged during DJF (Fig. 5c), MAM (Fig. 5e), and SON (Fig. 5i), and lowest during JJA (Fig. g, in fact, SFX-ROCK outperformed SFX-TEB over mixed surfaces during summer). The small differences in nighttime LST amongst simulations were also reflected in the S score, both on the annual (Fig. 5b) and seasonal scales (Figs. 5d, f, h, j).

SFX-TEB overestimated the observed daytime SUHI effect during MAM, JJA, and SON, while underestimating this effect during DJF (Fig. 6a). This resulted in an annual average overestimation of the SUHI intensity for this simulation. In contrast, ERA5 and SFX-ROCK largely underestimated the daytime SUHI effect over Paris (Fig. 6a). Indeed, misrepresentation of the urban radiative budget resulted in a nearly null SUHI effect in these simulations, as also illustrated by Fig. 2. One important result is that the magnitude of the daytime SUHI overestimation in SFX-TEB was smaller than the underestimation in ERA5 and SFX-ROCK, meaning that SFX-TEB improved the representation of the SUHI effect over all seasons (Fig. 6c). The statistical distribution of the daytime SUHI effect intensity was greatly improved in SFX-TEB compared to ERA5 and SFX-TEB throughout all seasons (Fig. 6e) - the annual average S score for the daytime SUHI PDF was 14% in ERA5, 27% in SFX-ROCK, and 81% in SFX-TEB.

During nighttime, the generally similar performance of all simulations over all types of surfaces resulted in a similar performance in reproducing the average nighttime UHI effect throughout all seasons (Fig. 6b). All simulations - ERA5, SFX-ROCK, and SFX-TEB – underestimated the observed nighttime Parisian SUHI, with the differences amongst simulations being within observational uncertainty (Fig. 6d). Nonetheless, the results revealed a better ability of the SFX-TEB in reproducing the nighttime SUHI statistics, which showed an annual average S score of 39%, clearly above the 11%





and 18% found for ERA5 and SFX-ROCK respectively (Fig. 6f). This improved representation of the nighttime SUHI

statistics was found during all seasons, being largest during JJA and lowest during DJF (Fig. 6f).

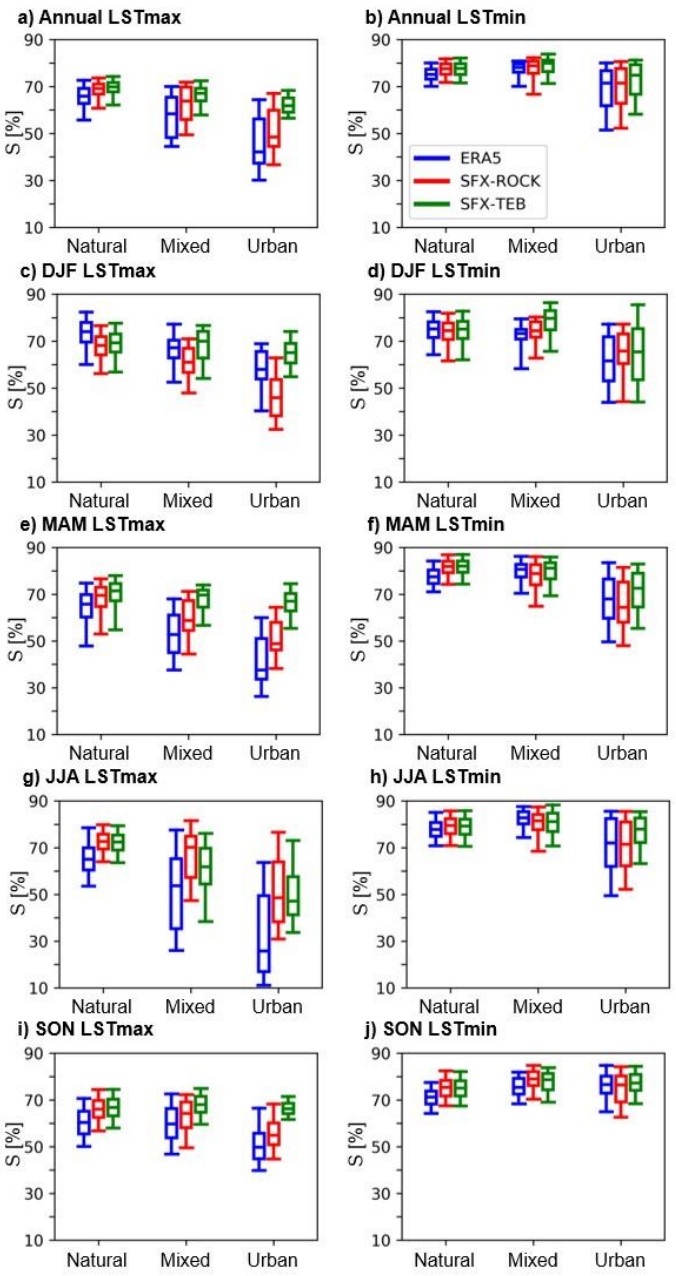

**Figure 5: Boxplots of S score computed over the 2004-2018 period from ERA5 (blue), SFX-ROCK (red), and SFX-TEB (green) for natural surfaces (grid boxes with urban fraction below 0.33), mixed surfaces (grid boxes with urban fraction between 0.33 and 0.66), and urban surfaces (grid boxes with urban fractions above 0.66). From top to bottom, the rows represent the S score computed for the full annual cycle, DJF, MAM, JJA, and SON.**



**Figure 6: Annual and seasonal average Parisian SUHI magnitude computed over the 2004-2018 period during a) daytime and b) nighttime. The corresponding annual and seasonal average |Bias| for daytime is represented in c) and for nighttime in d), and the corresponding annual and seasonal S score is represented in e) for daytime and f) for nighttime. The LSA-SAF is represented by black lines, ERA5 by blue markers, SFX-ROCK by red markers, and SFX-TEB by green markers.**

The observations show a relatively small range of the diurnal amplitude of the Parisian SUHI, which is largely overestimated by the SFX-TEB simulation (Fig. 7a). This reflects the overestimation of daytime LST and underestimation of nighttime LST over the urban grid boxes, as discussed above, in all seasons except winter (Fig. 7c). This issue is particularly pronounced during summer (Fig. 7g). Nonetheless, SFX-TEB still represented an improvement compared to ERA5 and





SFX-ROCK. On the one hand, ERA5 and SFX-ROCK failed to simulate any urban to rural contrast throughout the full annual cycle (Figs. 7a, c, e, g, and i). On the other hand, the annual average systematic errors in the simulated LST were lower in SFX-TEB compared to ERA5 and SFX-ROCK (Fig. 7b), although the differences only exceeded the observational uncertainty during daytime. The daytime reductions occurred mostly during DJF (Fig. 7d), MAM (Fig. 7f), and SON (Fig. 7j), while the daytime |Bias| values during JJA (Fig. 7h) and the nighttime |Bias| values during all seasons were within observational uncertainty.

### 3.2 Intercomparison of the simulated UHI over Paris

The $T_{2m}$ in ERA5 did not show any evidence of the UHI over Paris during daytime (Fig. 8a) nor during nighttime (Fig. 8b). The UHI effect did not emerge in SFX-ROCK during day- nor nighttime (respectively Fig. 8c and Fig. 8d). In contrast, the SFX-TEB $T_{2m}$ showed a signature of the UHI over Paris during daytime (Fig. 8e) and nighttime (Fig. 8f), in agreement with previous studies reporting the existence of the UHI effect over Paris (Lemonsu et al., 2014; Daniel et al., 2019). The enhanced performance in simulating the Paris UHI for SFX-TEB compared to ERA5 and SFX-ROCK was confirmed by the comparison against the UHI estimated from station observations (Fig. 9). During daytime, ERA5 and SFX-ROCK underestimated the observed Parisian UHI effect throughout all seasons, while SFX-TEB slightly overestimated the UHI (Fig. 9a), resulting in an overall reduction of the |Bias| in all seasons (Fig. 9c). On the annual average, the systematic error magnitude reduced from 0.7°C in ERA5 and SFX-ROCK to 0.2°C in SFX-TEB. Moreover, SFX-TEB also improved the statistics of the daily UHI magnitude throughout all seasons (Fig. 9e), resulting in an overall S score value of 87%, well above the 59% and 56% respectively, found for ERA5 and SFX-ROCK.

The results also showed that SFX-TEB improved the simulation of the UHI during nighttime when compared to ERA5 and SFX-ROCK (Fig. 9b), reducing the |Bias| error over all seasons by more than 1°C, resulting in an overall |Bias| value of 0.6°C in SFX-TEB, 2.5°C in ERA5, and 2.2°C in SFX-ROCK (Fig. 9d). SFX-TEB largely improved the representation of nighttime UHI statistical distribution in ERA5 and SFX-ROCK throughout all seasons (Fig. 9f). The overall S score was 25% for ERA5, 31% for SFX-ROCK, and 79% for SFX-TEB.

The annual average diurnal cycle of the Paris UHI clearly illustrated the contrasting results from ERA5 and SFX-ROCK to SFX-TEB. SFX-TEB showed an annual averaged UHI intensity varying between +1.9°C during the afternoon and night, and +0.1°C during the morning (Fig. 10a), whilst ERA5 and SFX-ROCK showed a nearly zero UHI effect throughout the entire diurnal cycle. The afternoon and nighttime UHI effect in SFX-TEB was highest during MAM (Fig. 10c) but remained above +1.5°C throughout all seasons (Fig. 10b-e), while the morning UHI was strongest during DJF (Fig. 10b) and its minimum reached close to zero values during MAM (Fig. 10c), JJA (Fig. 10d), and SON (Fig. 10e).

The relatively strong magnitude of the nighttime UHI in SFX-TEB corresponds to a significant improvement compared to ERA5 and SFX-ROCK. This result contrasts with the relatively small magnitude of the nighttime SUHI in SFX-TEB, which largely underestimates observations. The better performance of SFX-TEB in simulating the UHI compared to the SUHI was likely related to the ability of the model to represent part of the nighttime urban canopy layer heating associated with





prescribed anthropogenic heat fluxes, while the lack of land-atmosphere feedbacks inhibits this warmer canopy layer from affecting the LST.

The annual averaged SUHI measured as a difference between the urban and rural station locations under all-sky conditions computed from SFX-TEB peaked around 12h local time with a value of +5.0°C, and reduced throughout the afternoon stabilizing at a value around +1.5°C during the night and morning (Fig. 10f). The inability of ERA5 and SFX-ROCK to simulate the SUHI throughout the diurnal cycle was also clearly evidenced in Fig. 10f. The SUHI was strongest for all hours of the day during MAM (Fig. 10h) and JJA (Fig. 10i) and weakest during DJF (Fig. 10g) and SON (Fig. 10j). We highlight

the clear contrast between the UHI and SUHI diurnal cycles in Fig. 10. While the former peaked during late afternoon and night and decreased sharply during the morning, the latter showed opposite behavior peaking around midday and reaching the minimum during nighttime and late afternoon. This result was related to the thermal inertia of the canopy layer resulting in the well-known lag between surface and near-surface air warming.

However, the analysis of the annual averaged SUHI measured as a difference between the urban and rural station locations

under clear-sky conditions revealed significant differences in the average diurnal cycle simulated by SFX-TEB and observations (Fig. 10k). Indeed, SFX-TEB clearly overestimated the SUHI diurnal amplitude: the maximum amplitude of the SUHI average diurnal cycle was 2.9°C in observations and 6.8°C in SFX-TEB. This large overestimation resulted from SFX-TEB underestimation of the observed nighttime SUHI and overestimation during daytime. This effect was strongest during MAM (Fig. 10m) and JJA (Fig. 10n) and weakest during SON (Fig. 10o). During DJF (Fig. 10l), SFX-TEB underestimated

the observed clear-sky SUHI intensity throughout the entire diurnal cycle. Despite its limitations, SFX-TEB represents an improvement compared to ERA5 and SFX-ROCK which fail to simulate the Parisian SUHI effect throughout the entire diurnal cycle (Figs. 10 l-o).

The strong overestimation of the annual averaged daytime SUHI in SFX-TEB was associated with a sharp difference in the surface turbulent heat fluxes between the urban and rural station locations. The urban site showed lower average latent heat

flux (LH) values than the rural site by -75 Wm$^{-2}$ around noon (Fig. 10p) and higher sensible heat flux (SH) values by around +75 Wm$^{-2}$ (Fig. 10u). ERA5 and SFX-ROCK showed significantly smaller differences in LH and SH between urban and rural locations in agreement with their inability to simulate the UHI and SUHI. The differences in SH and LH between urban and rural locations in SFX-TEB were strongest during MAM (respectively Fig. 10r and 10w) and JJA (respectively Fig. 10s and 10x), and weaker during SON (respectively Fig. 10t and 10y) and DJF (respectively Fig. 10q and 10v). These results

suggest a direct link between the daytime overestimation of daytime SUHI in SFX-TEB with the lack of surface-atmosphere feedback in agreement with the recent findings of McNorton et al. (2021),

Finally, we noticed that the underestimation of the observed SUHI estimated as a point difference between the two station locations during night hours and the overestimation during morning and afternoon shown in Figs. 10k to 10o was coherent with the results presented in Fig. 6 for the daily minimum and daily maximum SUHI using the area-averaged SUHI

definition. Moreover, these results were also coherent with the overestimation of the daily maximum UHI and





underestimation of the daily minimum UHI presented in Fig. 9, including the daytime underestimation of both UHI and SUHI during DJF.

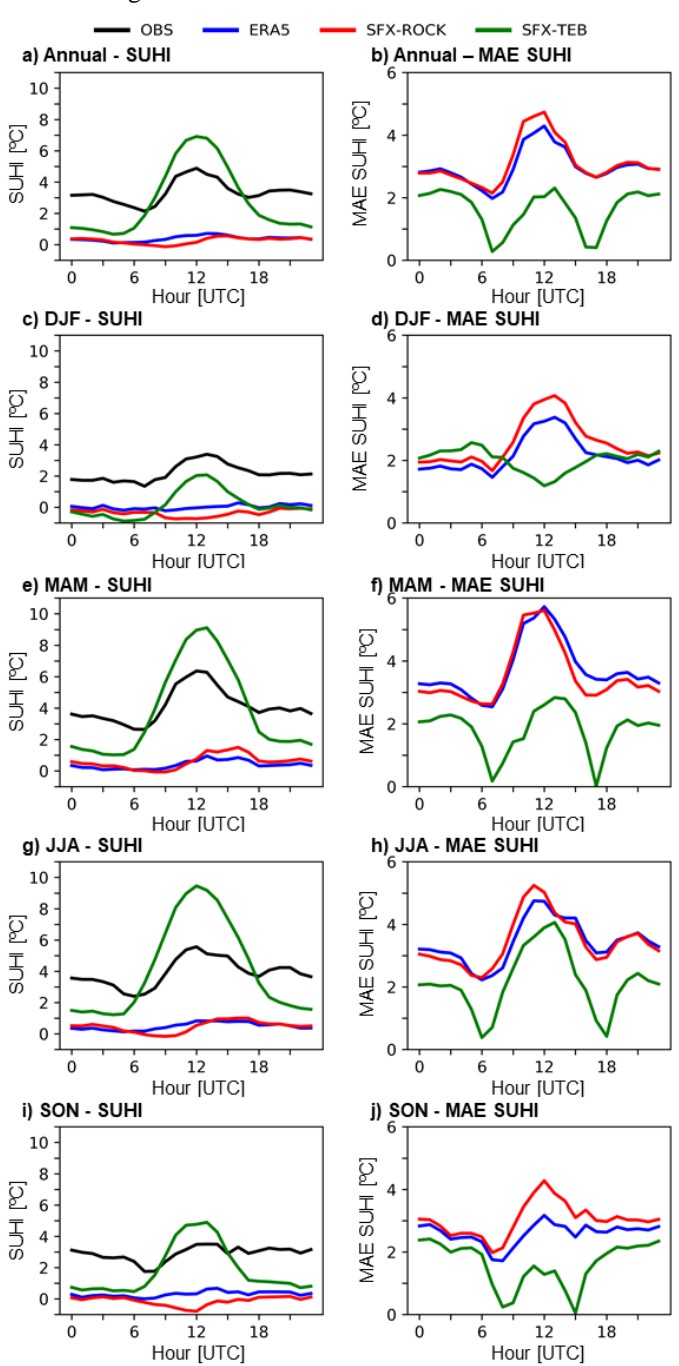

**Figure 7: Average SUHI diurnal cycle computed over the 2004-2018 period for a) full year, c) DJF, e) MAM, g) JJA, and i) SON. The different colors represent LSA-SAF (black), ERA5 (blue), SFX-ROCK (red), and SFX-TEB (green). The respective |Bias| diurnal cycles are represented in b) full year, d) DJF, f) MAM, h) JJA, and j) SON using LSA-SAF as reference.**





**Figure 8: $T_{2m}$ averaged over the 2004-2018 period during daytime (left column) and nighttime (right column) over the study domain computed from ERA5 (a) and b)), SFX-ROCK (c) and d)), and SFX-TEB (e) and f)).**







**Figure 9: Annual and seasonal average Parisian UHI magnitude computed over the 2004-2018 period during a) daytime and b) nighttime. The corresponding annual and seasonal average |Bias| for daytime is represented in c) and for nighttime in d), and the corresponding annual and seasonal S score is represented in e) for daytime and f) for nighttime. The observed UHI is represented by black lines, ERA5 by blue markers, SFX-ROCK by red markers, and SFX-TEB by green markers.**

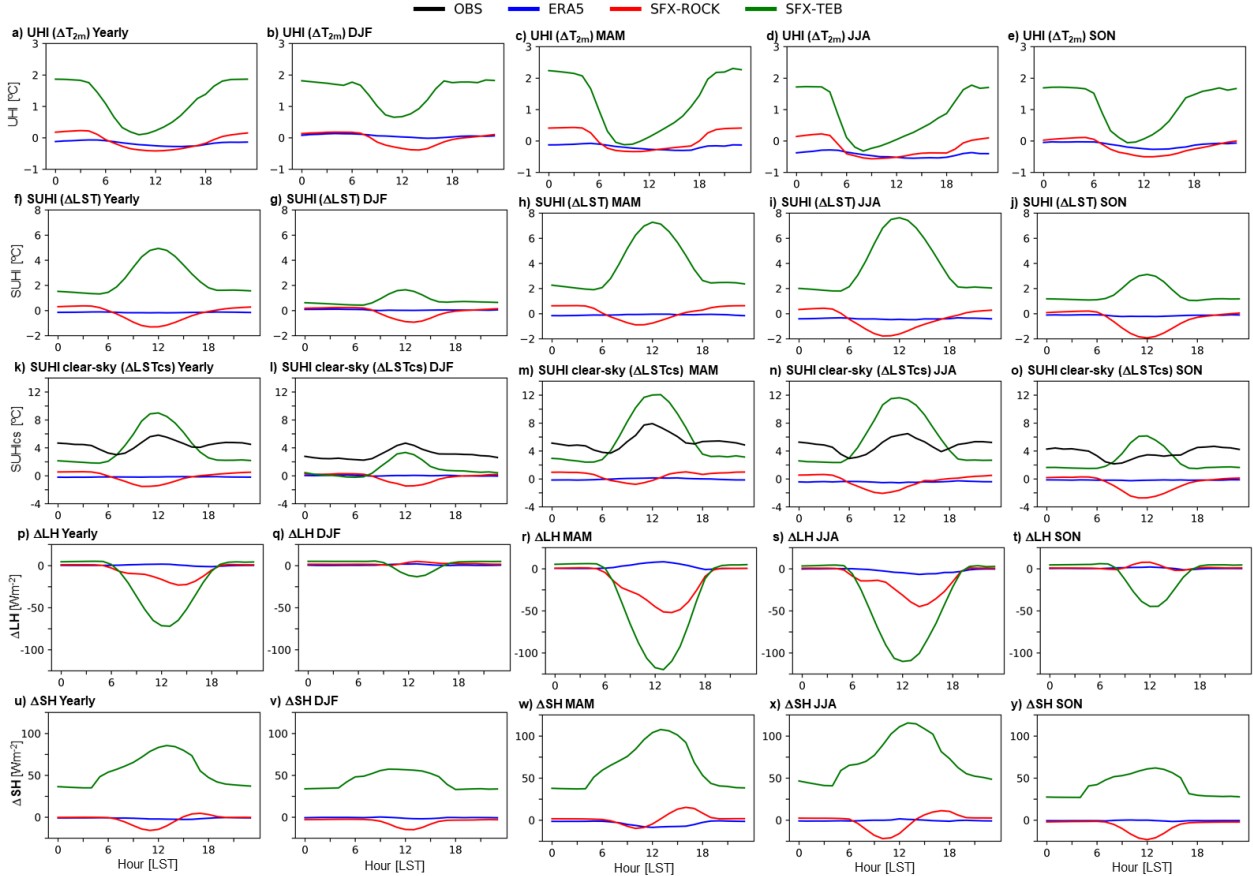

**Figure 10: Average diurnal cycle of UHI (top row), SUHI under all-sky conditions (second row), SUHI under clear-sky conditions (third row), ΔLH (difference between surface latent heat flux between urban and rural areas, fourth row), ΔSH (difference between surface sensible heat flux between urban and rural areas, last row). The columns from left-to-right represent averages taken over full year, DJF, MAM, JJA, and SON. The average diurnal cycles were computed over the 2004-2018 period. The different colors represent LSA-SAF (black), ERA5 (blue), SFX-ROCK (red), and SFX-TEB (green).**

## 4. Conclusions

We assessed the added value of the SURFEX offline simulations to downscale ERA5 reanalysis over urban areas, focusing on the urban heat island effect over the city of Paris. The relevance of this study is threefold. First, ERA5 is a widely used product for climate research and applications but an evaluation of its ability to represent urban climate is lacking. Second, it adds to recent works demonstrating the added value of the SURFEX offline downscaling framework for high-resolution computationally efficient urban climate simulation. Specifically, we leveraged on the hourly resolution of ERA5, the SURFEX simulations, and the LSA-SAF LST product to perform an unprecedented evaluation of the Parisian SUHI diurnal cycle, its key governing processes, and the ability of the different model setups to simulate them. Finally, the





intercomparison of long-duration high-spatial and temporal resolution datasets allow to explore the key sensitivities of the UHI and SUHI and their implications for the development of urban climate modelling.

Our results showed that ERA5 does not reproduce the observed UHI nor SUHI effects over Paris due to lack of representation of urban processes in the ECMWF IFS and, also, the relatively coarse grid resolution which is inappropriate for representing the highly heterogeneous urban environments. Increasing the grid resolution to the order of a few

kilometers or less was found to be an important but not sufficient condition for simulating the urban heat island. Indeed, the high-resolution simulation employing the bulk bare rock approach to urban climate parameterization commonly used in ESMs, GCMs and RCMs (see, e.g., Daniel et al., 2019; Langendijk et al., 2019; Zhao et al., 2021) was unable to reproduce the observed UHI and SUHI. In contrast, the high-resolution simulating employing an UCM for urban parameterization showed significant added value in reproducing the Parisian SUHI during daytime when compared to ERA5 and SFX-ROCK.

Specifically, SFX-TEB reduced the systematic errors of daily maximum LST over urban pixels for all seasons (yearly average reduction of 4.2ºC and 3.9ºC respectively compared to ERA5 and SFX-ROCK). Moreover, SFX-TEB also improved the simulation of the observed daytime LST PDF over urban areas during all seasons compared to ERA5 and SFX-ROCK (with S values up to 20% higher). On the other hand, the differences in $LST_{min}$ over urban surfaces and in $LST_{max}$ and $LST_{min}$ over natural surfaces were within observational uncertainty. In other words, SFX-TEB significantly improved the simulation

of urban daytime LST, while displaying similar skill to ERA5 and SFX-ROCK during nighttime over urban areas and, also, over natural surfaces throughout the entire diurnal cycle. It is important to note that uncertainties in remotely sensed LST due to its directional property should be negligible in this study due to the coarse spatial resolution of the SURFEX model.

SFX-TEB displayed the overall better performance in reproducing the observed daytime SUHI over Paris and its magnitude variability (with the respective S score increasing by 67% compared to ERA5). During nighttime, we found similar

performance amongst ERA5, SFX-ROCK and SFX-TEB in reproducing the nighttime SUHI effect throughout all seasons, with a generalized underestimation of the observed SUHI magnitude. Notice, however, that SFX-TEB showed a slight improvement in simulating the nighttime SUHI spatial pattern and temporal variability compared to ERA5 and SFX-ROCK.

ERA5 and SFX-ROCK did not reproduce the Parisian UHI effect throughout the entire diurnal cycle. In contrast, SFX-TEB displayed significant added value in simulating the observed UHI during daytime and nighttime, resulting in an annual

average bias magnitude reduction of 0.5ºC for daytime and more than 1.5ºC for nighttime compared to ERA5 and SFX-ROCK. The distribution of daily UHI variability was also improved in SFX-TEB, with the S score increased by roughly 30% for daytime and 50% for nighttime. The improved nighttime performance of SFX-TEB in simulating UHI relative to the SUHI may be explained by the ability of SFX-TEB to warm the urban canopy layer by anthropogenic heat releases combined with the lack of land-atmosphere feedbacks, which inhibits an LST response to the nighttime UHI. Finally, an

analysis of the diurnal cycle of the simulated surface turbulent heat fluxes suggests that SFX-TEB overestimates the urban/rural contrasts in SH and LH during daytime (particularly during warmer months). However, ERA5 and SFX-ROCK consistently underestimate the $T_{2m}$, LST and turbulent flux contrast throughout the entire diurnal cycle.



In summary, we highlight the large potential of the offline SURFEX-TEB framework for urban climate projections given its ability to produce computationally efficient high-resolution climate projections with increased accuracy compared to state-
of-the-art ESMs, GCMs, and RCMs. Moreover, its relatively small computational cost allows to perform a large number of climate experiments to investigate the impact of city-scale climate adaptation and mitigation strategies under different future emission scenarios. This framework may be improved in the future by including simplified representations of key land-atmospheric feedbacks. Specifically, the possibility to improve the ability of this framework in reproducing the nighttime SUHI by including coherent dynamical corrections to the forcing temperature and downwelling longwave radiation fields
based on the $T_{2m}$ diagnostic in the previous timestep will be investigated in a subsequent work.

**Code availability**

The SURFEX modeling platform of Météo-France is open source and can be downloaded freely at http://www.umr-cnrm.fr/surfex/ (CNRM, 2016). It uses the CECILL-C license, a French equivalent to the L-GPL license (http://cecill.info/licences/Licence_CeCILL_V1.1-US.html; CEA-CNRS-Inria, 2013). It is updated at a relatively low
frequency (every 3 to 6 months). If more frequent updates are needed – or if what is required is not in Open-SURFEX (DrHOOK, FA/LFI formats or GAUSSIAN grid) – you are invited to follow the procedure to get an SVN account and to access real-time modifications of the code (see the instructions in the first link). In this study, SURFEX's version 8.1 was used.

**Data availability**

ERA5 data can be obtained freely from the Copernicus Climate Change Service Information website (https://climate.copernicus.eu/, Copernicus Climate Change Service (C3S), 2019).
The LSA-SAF LST can be obtained freely from their website (https://landsaf.ipma.pt/, last access: 13 December 2021).
The NCDC CSOD weather station data can be obtained from their website (https://www.ncei.noaa.gov/access/metadata/landing-page/bin/iso?id=gov.noaa.ncdc:C00516, last access: 14 December
2021).
The considered fields (LST, T2m) from the simulations (SFX-TEB, SFX-ROCK) considered in the present study are freely available at https://doi.org/10.5281/zenodo.5780448 (Nogueira et al., 2021).





## Author contributions

Conceptualization was done by MN, PMS and ED. LST data was acquired and processed by AH and SE. ERA5 processing
was carried out by DL. Formal analysis and the original draft writing were done by MN. All authors contributed to writing,
reviewing, and editing the paper.

## Competing interests

The authors declare that they have no conflict of interest.

## Acknowledgments

The authors wish to acknowledge the CONTROL and LEADING projects, both funded by FCT. Daniela C.A. Lima was
funded by FCT under project LEADING (PTDC/CTA-MET/28914/2017). Pedro M.M. Soares would like to acknowledge
the financial support of FCT through project UIDB/50019/2020 – IDL and EEA-Financial Mechanism 2014-2021 and the
Portuguese Environment Agency through Pre-defined Project-2 National Roadmap for Adaptation XXI (PDP-2). Frederico
Johannsen was funded by FCT under the research grant UI/BD/151498/2021. The authors would also like to acknowledge
the financial support by FCT through project UIDB/50019/2020 – Instituto Dom Luiz. Finally, all authors acknowledge the
ECMWF for producing ERA5.

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
