# Peer review of "Assessment of the Paris urban heat island in ERA5 and offline SURFEX-TEB (v8.1) simulations using METEOSAT land surface temperature product"

_Geoscientific Model Development, 2021_

## Referee Comment (RC2)

Review for Manuscript GMD Manuscript entitled:
*Assessment of the Paris urban heat island in ERA5 and offline SURFEX-TEB (v8.1) simulations using METEOSAT land surface temperature product*

**Summary:**
The authors use offline urban climate simulations and ERA-5 reanalysis data to demonstrate that the ERA-5 dataset does not accurately represent the spatial pattern of built-environment induced warming. They conduct two sets of simulations, based on the premise that "bulk urban parameterization is often employed in state-of-the-art regional climate simulations", replacing the urban landscape with a bulk "rock covers" for one set, and using the TEB UCM for the second set of simulations.

The science is reasonable, the analysis is sound, but the justification for the work done is disingenuous at best, or patently false at worst. To state that "bulk urban parameterization is often employed in state-of-the-art regional climate simulations" couldn't be further from the truth. UCM models of varied complexity are used the world over and these are detailed below.

Although critically important, because the current language considerably misconstrues the importance of the research, I do not view the required modifications to the manuscript to be major and do believe that once the language is toned down, the paper will become suitable for publication.

**Specific comments**
**Abstract:** However, most of the state-of-the-art global and regional climate models have an oversimplified representation of (or completely neglect) urban climate processes.
**Comment:** This is certainly not the case for "most regional climate models" and details are provided below. This cannot be used as a justification for the work provided since "most" RCMs and urban climate modeling researchers have been using a varied complexity of single or multi-layer urban canopy models for the last 1-2 decades!

**Abstract:** Finally, the offline SURFEX-TEB framework applied here demonstrates the ability to simulate the urban climate, which is an asset to build urban climate projections that allow the development of mitigation and adaptation strategies.
**Comment:** improved characterization of urban climate change requires coupled simulation, rather than offline simulation where the built environment is forced by the overlying atmosphere but does not have a chance to interact with it. If the objective is to "build urban climate projections that allow the development of mitigation and adaptation strategies" it becomes difficult to justify why offline simulations are the way to go.

**Line 70:** Moreover, while observations cover the past, numerical simulations can be extended to the future and, therefore, consider different scenarios of future socio-economic evolution, urban development, and adaptation strategies.

**Comment:** Indeed, this is an excellent point and requires supporting references since such work is now increasingly performed. For example, the very first large-scale effort to conduct such mesoscale (process-based) coupled (urban UCM to the overlying atmosphere) simulations, accounting for both urban expansion and greenhouse gas induced climate change (i.e., socioeconomic evolution), allowing for a direct comparison among the urban environment forcing agents, including adaptation strategies that may offset this warming, should provide context for the readership and should be acknowledged:

Georgescu, M., Morefield, P. E., Bierwagen, B. G., & Weaver, C. P. (2014). Urban adaptation can roll back warming of emerging megapolitan regions. Proceedings of the National Academy of Sciences, 111(8), 2909-2914

**Line 72:** Most state-of-the-art climate models do not consider or have simplified representations of the urban environments (Garuma, 2018; Zhao et al., 2021).

**Comment:** It is at this point that the language begins to require a toning down, as it completely misrepresents the state-of-the-science.

For example, many RCMs today (and for the last decade or two) widely use various versions of a single layer urban canopy model that accounts for the building geometry of cities, shadowing from buildings, and even anthropogenic heating (e.g., Kusaka et al., 2001):

Kusaka H, Kondo H, Kikegawa Y and Kimura F 2001 A simple single-layer urban canopy model for atmospheric models: comparison with multi-layer and slab models Bound.-Layer Meteorol. 101 329–58.

In addition, more complex models (multi-layer) are increasingly being used in such process-based models. The wording here (and elsewhere) needs to be modified as it is not an accurate reflection of the state-of-the-science (in the Abstract as well and other locations as needed).

**Line 82:** However, the use of UCM coupled to RCMs is not a standard procedure for climate simulation (and is not projected to be in the next generation of multi-model RCM ensembles) due to its very high computational costs, resulting in a poor representation of many aspects of urban climate in state-of-the-art RCM ensemble datasets

**Comment:** Again, RCMs coupled with UCMs have been used in climate mode. While this is not "standard", the statement is misleading since it omits the realization of such simulations (e.g., Krayenhoff et al., 2018, which was actually referenced in this manuscript).

**Line 91:** Despite those limitations, recent studies have demonstrated the added value of this approach in reproducing key features of observed urban climate compared to traditional climate simulations (without representation of urban processes), including the UHI and the frequency, intensity and duration of urban extreme temperature events (Broadbent et al., 2018; …)

**Comment:** Neglecting advective processes within the urban climate modeling domain is a major shortcoming. The "added value" referred to by the authors is simply a reference to reduced computational limitations, but nothing more. The question then becomes "What is the value lost" when using this simplified approach? To my knowledge, this question has not been addressed. Again, the justification for the performed work relies on an assessment of the science that is roughly 2 decades old.

**Line 97:** … demonstrated how this type of framework may be used to disentangle the impact of land-use change, from large-scale warming induced by greenhouse gas emissions, and from natural climate variability.

**Comment:** Again, this has been done using traditional dynamical downscaling for climate simulations (decadal length simulations) - I feel it is a little disingenuous to omit this work since considerable research in this area to "disentangle the impact of land-use change from large-scale warming induced by greenhouse gas emissions" has already been performed (e.g., Georgescu et al., 2014; Krayenhoff et al, 2018; Broadbent et al.; 2020). The issue of "natural climate variability is certainly an additional distinction and that requires simulations on the order of many decades or longer (e.g., AMO or PDO cycles):
Broadbent, A. M., Krayenhoff, E. S., & Georgescu, M. (2020). The motley drivers of heat and cold exposure in 21st century US cities. Proceedings of the National Academy of Sciences, 117(35), 21108-21117

Line 145: "last-generation"
**Comment:** Change to "latest-generation".

Line 173: This bulk urban parameterization is often employed in state-of-the-art regional climate simulations.
**Comment:** This is oce again patently false, as detailed with the already specified references, and should be removed. Many scientists are now using multi-layer schemes, and some are even using building energy models coupled to the multi-layer scheme (e.g., see several of Francisco Salamanca's papers and please perform a thorough read of Fei Chen's 2011 Int. J. Clim. paper that goes deep into the issue of varied urban parameterizations beyond the bulk scheme, many of which, as already mentioned and referenced in my comments, have been used for 1-2 decades):
Salamanca, F., Krpo, A., Martilli, A., & Clappier, A. (2010). A new building energy model coupled with an urban canopy parameterization for urban climate simulations—part I. formulation, verification, and sensitivity analysis of the model. *Theoretical and applied climatology*, *99*(3), 331-344

Salamanca et al 2011 compared WRF performance using a bulk scheme to more advanced urban representations more than 1 decade ago:

Salamanca, F., Martilli, A., Tewari, M., & Chen, F. (2011). A study of the urban boundary layer using different urban parameterizations and high-resolution urban canopy parameters with WRF. *Journal of Applied Meteorology and Climatology*, *50*(5), 1107-1128

---

## Author Comment (AC1)

We want to thank the editor and both anonymous referees for the constructive comments that helped improve our article. Below, we provide our detailed replies to the comments of the two anonymous referees.

**Anonymous Referee #1**

**The present article evaluates the Paris Surface Urban Heat Island (SUHI) and Urban Heat Island (UHI) in the ERA5 re-analysis data and Offline simulations with the land surface model SURFEX using a "rock-type" urban parametrisation or the urban canopy model TEB to simulate the urban surface energy balance.**
**Observed SUHI is derived from LSA-SAF (SEVIRI) satellite observations, UHI from station data.**
**Results show that ERA5 does not capture the SUHI or UHI, which is no surprise given that the underlying IFS model has not been using an urban parametrisation and no urban stations have been assimilated.**
**A major improvement of the results for SUHI and UHI is found when using SURFEX-TEB.**
**This shows that even with an Offline application of SURFEX-TEB, re-analysis products like ERA5 could be strongly improved and spatially refined in urbanised areas, which is a very useful finding.**

**The present paper is interesting, very well structured and written, the results are clear, so I have only very minor comments:**

R: We are grateful for the reviewer's positive statements about the paper, and for spending their time analysing and reviewing our article. Below, we present our replies to each comment.

1) **There is a recent study on the Paris urban climate including the SUHI: Benjamin Le Roy, Aude Lemonsu, Raphaëlle Kounkoud-Arnaud, Denis Brion, Valéry Masson: Long time series spatialized data for urban climatological studies: a case study of Paris, France. International Journal of Climatology, Wiley, 2019, ï¿¿10.1002/joc.6414ï¿¿.**
**The results from the present study should be compared with this study.**

R: We agree with the reviewer's comment and therefore we added at line 270 (Results section, when presenting the seasonal SUHI) the following statement: "At a seasonal scale the observed SUHI reveals a clear seasonal cycle at daytime and a rather constant value at nighttime (Figs. 6a, b). At daytime (nighttime), the winter and summer SUHI effect amounts to around 3 and 6 ºC (2 and 2.5 ºC), respectively. UHI presents a less pronounced seasonal cycle at daytime and similar values at nighttime in relation to SUHI (Figs. 9a, b). At daytime (nighttime), the

winter and summer UHI effect amounts to around 0.6 and 0.25 ºC (2 and 2.75 ºC), respectively. These results are similar to what Roy et al. (2020) found when studied the intensity and spatial extent of the UHI and SUHI over Paris using 1-km resolution observational datasets. Both studies agree in the SUHI and UHI annual cycle, although displaying some differences in their intensities, namely in daytime SUHI (4 ºC in summer, 2 ºC in winter) and nighttime UHI (2 ºC in summer, 1 ºC in winter) (Fig. 9b), both more intense in our study. These differences may arise from a number of reasons: the temporal ranges considered were different (2004-2018 in our study vs 2000-2016); Roy et al. (2020) considered a much larger rural area and the LST satellite data was retrieved from MODIS (which has higher spatial resolution than SEVIRI but at the cost of lower temporal resolution, with only two daily observations); finally, the T2M observations were generated from a gridded dataset developed at Météo-France while ours were obtained directly from two in-situ weather stations."

2) **The absolute value of the bias (|Bias|) is used throughout the paper. I dont understand why this is done, since the information on the sign of the bias is lost. I propose to use the Bias as is, and change the related figures and text.**

R: We changed Figures 3,6,9 and the text accordingly. We kept the MAE in Figure 7 for visual purposes.

**L114: I am wondering how to simulate the UHI with a single-column approach. In fact, the advection of cool air from the rural areas around the city towards the city cannot be taken into account. Do these simulations parametrise the advection via a forcing term?**

R: We fully understand the reviewer's comment. The SURFEX-TEB does not account for advection and we agree with the need for such a process representation. In fact, we are working on that kind of improvement, but it is still under development. And we focus those future developments in the Conclusions.

**L121: I guess all observations are for the same 2004-2018 period?**
**It seems not to be stated explicitly for the station observations.**

R: Yes, the observations are all from the 2004-2018 period. We added this detail in the first paragraph of the Observations and Reanalysis subsection.

**L177: Schoetter et al. (2020) has only dealt with simulations on Hong Kong, so did not formally show that the single-layer TEB is adequate for mid-rise cities. So I think this statement has to be changed.**

R: Thanks for the hint. We revisited the Schoetter et al. (2020) study and they refer that the single-layer TEB is suitable for mid-rise cities and also cite Trusilova et al. (2016), therefore, we also added the latter reference to the statement and to the list of references. Trusilova, K., Schubert, S., Wouters, H., Früh, B., Grossman-Clarke, S., Demuzere, M., and Becker, P.: The urban land use in the COSMO-CLM model: a comparison of three parameterizations for Berlin, 25, 231–244, https://doi.org/10.1127/metz/2015/0587, 2016.

**Figure 2: The comparison is a bit unfair since the ERA5 resolution is so coarse. You could add a third column with the other datasets interpolated to the ERA5 grid.**

R: We understand the reviewer's comment. In Figure 2, we do not aim at a direct quantitative comparison with ERA-5 but instead we want to illustrate that the city urban effect is nearly absent.

**Figures 3, 5 and 6: More space should be added between the different lines of figures.**

R: We followed the suggestion.

**Figures 6 and 9: In the legend you should use points instead of lines for ERA5, SFX-ROCK, SFX-TEB.**

R: Thank you. Corrected.

**Figure 8ace: There are some weird features (rectangle-shaped). What is their origin?**

R: Those features are a signature of the coarser resolution of the forcing (0.25º) combined with the strong constraint of the T2M simulated by the SURFEX offline scheme on the atmospheric forcing. For LST, this constraint is smaller (depends more on land surface processes explicitly simulated by the land surface model). When a more different urban parameterization is used (SFX-TEB) then SURFEX has a stronger effect in modulating T2M, hence the rectangle-shaped features are less pronounced.

**L403: high-resolution simulation.**

R: Corrected.

**Anonymous Referee #2**

**Summary:**

**The authors use offline urban climate simulations and ERA-5 reanalysis data to demonstrate that the ERA-5 dataset does not accurately represent the spatial pattern of built-environment induced warming. They conduct two sets of simulations, based on the premise that "bulk urban parameterization is often employed in state-of-the-art regional climate simulations", replacing the urban landscape with a bulk "rock covers" for one set, and using the TEB UCM for the second set of simulations.**

**The science is reasonable, the analysis is sound, but the justification for the work done is disingenuous at best, or patently false at worst. To state that "bulk urban parameterization is often employed in state-of-the-art regional climate simulations" couldn't be further from the truth. UCM models of varied complexity are used the world over and these are detailed below.**

**Although critically important, because the current language considerably misconstrues the importance of the research, I do not view the required modifications to the manuscript to be major and do believe that once the language is toned down, the paper will become suitable for publication.**

R: We would like to thank the reviewer for their insightful and constructive comments that helped to greatly improve our manuscript. Below, we reply to each comment separately.

**Specific comments**

**Abstract: However, most of the state-of-the-art global and regional climate models have an oversimplified representation of (or completely neglect) urban climate processes.**

**Comment: This is certainly not the case for "most regional climate models" and details are provided below. This cannot be used as a justification for the work provided since "most" RCMs and urban climate modeling researchers have been using a varied complexity of single or multi-layer urban canopy models for the last 1-2 decades!**

R: Thank you for your comment. You are right about the availability of complex urban parameterizations in modelling systems. Here, we were referring to its use in large simulation ensembles of Earth System (Global) Climate Models and Regional Climate Models such as CMIP5/6 and CORDEX (e.g. Zhao et al., 2021; Table A1 in Langendijk et al., 2019). In fact, these large ensemble efforts correspond to very long climate runs, of the order of one century (and in some cases a few decades).

For these climate scales, to the best of our knowledge, there are no consistent large scale ensembles produced using in a systematic way more complex urban schemes.

Therefore, we replaced the aforementioned statement in the abstract with "However, most of the large ensembles of global and regional climate model simulations do not include sophisticated urban parameterizations."

Throughout the manuscript we changed the statements where these differences between models and large ensemble simulations were not clear, namely in the Introduction and Conclusions, specifying them accordingly whenever mentioned in the comments that follow. We want to make clear that this distinction is highly relevant, and we thank the reviewer again for the comment.

**Abstract: Finally, the offline SURFEX-TEB framework applied here demonstrates the ability to simulate the urban climate, which is an asset to build urban climate projections that allow the development of mitigation and adaptation strategies.**

**Comment: improved characterization of urban climate change requires coupled simulation, rather than offline simulation where the built environment is forced by the overlying atmosphere but does not have a chance to interact with it. If the objective is to "build urban climate projections that allow the development of mitigation and adaptation strategies" it becomes difficult to justify why offline simulations are the way to go.**

R: We fully agree with the reviewer that a 3D coupled simulation view of cities is the future for urban climate modelling. However, we believe that in the forthcoming years, if not decade, it is not foreseen for the scientific community to be able to produce large climate simulations for climate change assessment studies, at very high resolutions, as required for that kind of urban simulations. Our goal here is to show the added value of the methodology proposed in this study for already available climate ensembles that still mostly use bulk urban schemes.

We rephrased the last statement of the Abstract to: "Finally, the offline SURFEX-TEB framework applied here demonstrates the added value of using more comprehensive urban parameterizations to simulate the urban climate, therefore, improving urban climate projections."

**Line 70: Moreover, while observations cover the past, numerical simulations can be extended to the future and, therefore, consider different scenarios of future socio-economic evolution, urban development, and adaptation strategies.**

**Comment: Indeed, this is an excellent point and requires supporting references since such work is now increasingly performed. For example, the very first large-scale effort to conduct such mesoscale (process-based) coupled (urban UCM to the overlying atmosphere) simulations, accounting for both urban expansion and greenhouse gas induced climate change (i.e., socioeconomic evolution), allowing for a direct comparison among the urban environment forcing agents, including adaptation strategies that may offset this warming, should provide context for the readership and should be acknowledged: Georgescu, M., Morefield, P. E., Bierwagen, B. G., & Weaver, C. P. (2014). Urban adaptation can roll back warming of emerging megapolitan regions. Proceedings of the National Academy of Sciences, 111(8), 2909-2914**

R: Thank you for the suggestion. We included in our introduction and discussion a relevant mention to this study.

We added to the statement in Line 69: "(...) numerical simulations can be extended to the future and, therefore, consider different scenarios of future socio-economic evolution, urban development, and adaptation strategies, as shown, for example, in Georgescu et al. (2014) where it was shown how urban planning could help offset the global warming effect in U.S. cities in the future."

**Line 72: Most state-of-the-art climate models do not consider or have simplified representations of the urban environments (Garuma, 2018; Zhao et al., 2021).**

**Comment: It is at this point that the language begins to require a toning down, as it completely misrepresents the state-of-the-science.**

**For example, many RCMs today (and for the last decade or two) widely use various versions of a single layer urban canopy model that accounts for the building geometry of cities, shadowing from buildings, and even anthropogenic heating (e.g., Kusaka et al., 2001): Kusaka H, Kondo H, Kikegawa Y and Kimura F 2001 A simple single-layer urban canopy model for atmospheric models: comparison with multi-layer and slab models Bound.-Layer Meteorol. 101 329–58.**

**In addition, more complex models (multi-layer) are increasingly being used in such process-based models. The wording here (and elsewhere) needs to be modified as it is not an accurate reflection of the state-of-the-science (in the Abstract as well and other locations as needed).**

R: You are completely right, these representations are present in many sophisticated models and have been used in short-term localized simulations and

case studies, like the article you mentioned. However, they have not been used in the generation of global and regional climate simulations such as CMPI5/6 and CORDEX, as already mentioned.

We changed the statement at line 73 to: "Most large ensembles of global and regional climate model simulations have simplified representations of the urban environment (Garuma, 2018; Zhao et al., 2021)."

**Line 82: However, the use of UCM coupled to RCMs is not a standard procedure for climate simulation (and is not projected to be in the next generation of multi-model RCM ensembles) due to its very high computational costs, resulting in a poor representation of many aspects of urban climate in state-of-the-art RCM ensemble datasets**

**Comment: Again, RCMs coupled with UCMs have been used in climate mode. While this is not "standard", the statement is misleading since it omits the realization of such simulations (e.g., Krayenhoff et al., 2018, which was actually referenced in this manuscript).**

R: Thank you for the heads-up comment. We changed the previous statement to: "However, the use of UCM coupled to RCMs is not a standard procedure for long-time/century climate simulations (and is not projected to be in the next generation of multi-model RCM ensembles) due to its very high computational costs, resulting in a poor representation of many aspects of urban climate in those RCM ensemble datasets."

**Line 91: Despite those limitations, recent studies have demonstrated the added value of this approach in reproducing key features of observed urban climate compared to traditional climate simulations (without representation of urban processes), including the UHI and the frequency, intensity and duration of urban extreme temperature events (Broadbent et al., 2018; ...)**

**Comment: Neglecting advective processes within the urban climate modeling domain is a major shortcoming. The "added value" referred to by the authors is simply a reference to reduced computational limitations, but nothing more. The question then becomes "What is the value lost" when using this simplified approach? To my knowledge, this question has not been addressed. Again, the justification for the performed work relies on an assessment of the science that is roughly 2 decades old.**

R: We agree that advective processes are very important and the atmosphere-surface coupling is highly relevant. However, our results show a clear added value

of using much less computationally demanding offline simulations when compared with the regional climate simulations output. It is important to keep in mind that in present days the only way of generating a large-scale ensemble of urban simulations is performing offline simulations for climate change assessments. Additionally, this approach allows the study of urban adaptation measures in a wider setting perspective as shown by Nogueira & Soares (2019).

**Line 97: … demonstrated how this type of framework may be used to disentangle the impact of land-use change, from large-scale warming induced by greenhouse gas emissions, and from natural climate variability.**

**Comment: Again, this has been done using traditional dynamical downscaling for climate simulations (decadal length simulations) - I feel it is a little disingenuous to omit this work since considerable research in this area to "disentangle the impact of land-use change from large-scale warming induced by greenhouse gas emissions" has already been performed (e.g., Georgescu et al., 2014; Krayenhoff et al, 2018; Broadbent et al.; 2020). The issue of "natural climate variability is certainly an additional distinction and that requires simulations on the order of many decades or longer (e.g., AMO or PDO cycles): Broadbent, A. M., Krayenhoff, E. S., & Georgescu, M. (2020). The motley drivers of heat and cold exposure in 21st century US cities. Proceedings of the National Academy of Sciences, 117(35), 21108-21117**

R: We proposed this approach in Nogueira & Soares (2019). Other approaches have been suggested in the past, of course. We acknowledged the suggested articles in our text.

Line 98: "Additionally, Nogueira and Soares (2019) demonstrated how this type of framework may be used to disentangle the impact of land-use change, from large-scale warming induced by greenhouse gas emissions, and from natural climate variability. Other approaches to tackle this problem have been suggested in the past (e.g. Georgescu et al., 2014; Krayenhoff et al., 2018; Broadbent et al., 2020)."

**Line 145: "last-generation"**

**Comment: Change to "latest-generation".**

R: Changed.

**Line 173: This bulk urban parameterization is often employed in state-of-the-art regional climate simulations.**

**Comment: This is oce again patently false, as detailed with the already specified references, and should be removed. Many scientists are now using multi-layer schemes, and some are even using building energy models coupled to the multi-layer scheme (e.g., see several of Francisco Salamanca's papers and please perform a thorough read of Fei Chen's 2011 Int. J. Clim. paper that goes deep into the issue of varied urban parameterizations beyond the bulk scheme, many of which, as already mentioned and referenced in my comments, have been used for 1-2 decades): Salamanca, F., Krpo, A., Martilli, A., & Clappier, A. (2010). A new building energy model coupled with an urban canopy parameterization for urban climate simulations—part I. formulation, verification, and sensitivity analysis of the model. Theoretical and applied climatology, 99(3), 331-344**

**Salamanca et al 2011 compared WRF performance using a bulk scheme to more advanced urban representations more than 1 decade ago: Salamanca, F., Martilli, A., Tewari, M., & Chen, F. (2011). A study of the urban boundary layer using different urban parameterizations and high-resolution urban canopy parameters with WRF. Journal of Applied Meteorology and Climatology, 50(5), 1107-1128**

R: Thank you for the suggested articles. We rephrased the aforementioned sentence, and we acknowledged these studies in the following statement that was introduced in line 170: "This bulk urban parameterization is often employed in large ensembles of regional climate simulations (…) It is worth pointing out, however, that several studies presenting RCMs combined with more complex urban schemes in short-term case studies have previously shown added value in simulating urban climate (e.g. Salamanca et al., 2010; Salamanca et al., 2011)."

---

## Author Response (AR2)

We want to thank the editor and the anonymous referee for their positive and constructive comments. Below, we provide our point-by-point reply.

**Anonymous Referee #1**

**The authors have well taken into account the reviewers comments and I have only some minor remarks. The Line numbers refer the tracked changes version of the manuscript.**
R: We are grateful for the reviewer's positive statements about the paper. We provide below our replies to each comment.

**Line 12: You might specify which ensembles of RCM simulations (e.g. EURO-CORDEX); such that the sentence becomes more precise.**
R: We added EURO-CORDEX and CMIP5/6 as examples of RCM and GCM ensembles.

**Line 72: "shown" is used twice in the same sentence.**
R: Second "shown" replaced with "demonstrated".

**Line 103: "other approaches": this is not very precise, so please provide a bit more details on what has actually been done.**
R: We agree with the reviewer, we provided a more detailed statement in Line 103: "Other approaches to tackle this problem have been suggested in the past, namely using dynamical downscaling to run climate simulations at the start and at the end of the century (e.g. Georgescu et al., 2014; Krayenhoff et al., 2018; Broadbent et al., 2020)."

**L176: The order of references is wrong. It should be the oldest first.**
R: Fixed.

**L331: Must it not be "while also changing its sign"?**
R: Thank you. Fixed.

**Topical Editor**

**Please check if the comments below are relevant to provide more synthetic information to improve your manuscript readability.**

**1. L38-43:**
**- Could you replace Oke (1982) or add to more updated reference, Oke et al. (2017, Urban Climates, T. R. Oke, H. Mills, A. Christen, Cambridge Univ. Press)**
**- There are studies to relate long-term UHI to economic conditions and add them to this paragraph (e.g., Hong et al., 2019; Li et al., 2020; He et al., 2022).**

**Hong et al. (2019) Temporal dynamics of urban heat island correlated with the socio-economic development over the past half-century in Seoul, Korea, Environmental Pollution.**

**Li et al. (2020) Socioeconomic drivers of urban heat island effect: Empirical evidence from major Chinese cities, Sustainable Cities and Society**

**He et al. (2022) Localized synergies between heat waves and urban heat islands: Implications to human thermal comfort and urban heat management, Environmental Research**

R: We thank the editor for providing these relevant references which were promptly added to the paragraph and to the list of references at the end of the manuscript.

**2. L88-90: I expect that UCM considers vertical exchanges of energy and mass only, which may emphasize the importance of your study.**
R: UCMs when run in offline mode neglect the feedbacks between the urban canopy layer and the air above (that is, the atmospheric forcing of the model). Technically, it does include some feedbacks, namely between the land surface and the urban canopy layer.

**3. L 139: Could you also mention uncertainty of LST by building material emissivity?**
R: We added it to the sentence in Line 139.

**4. L174-176: I am quite sure if this sentence is in the relevant position when I consider the sentences before this one.**

R: The sentence was moved to the Introduction section in Line 86.

**5. Section 2.3: For comparison of other studies, I recommend that you provide RMSE in the manuscript.**

R: Added a new figure for RMSE (equivalent to Figures 3 and 5 but for RMSE), which is the new Figure 6. The old Figures 6,7,8,9,10 are now Figures 7,8,9,10,11. RMSE was also included in Figures 7 and 10 (old Figures 6 and 9). The Results and Discussion section was updated to include RMSE in the text.

**6. Figure 3:**

**- Can you reduce negative bias in the SFX-ROCK experiment by modifying thermal properties of rock covers in the experiment? This may provide some hints for the simple diagnostic values for UHI simulations.**

R: It is likely that modifying the thermal properties of rock covers could change the errors. However, this would require calibration. Instead, here we are using the standard SFX-ROCK land cover, compared to using a more realistic representation of the urban cover using the SFX-TEB scheme, which also includes a more realistic representation of the surface thermal properties obtained from ECOCLIMAP database.

**- Also, I wonder if you can discuss why SFX-TEB shows substantial positive bias of daytime LSTmax in JJA. I wonder how to assign anthropogenic heat emission in summer and winter.**

R: The substantial positive bias of daytime LSTmax present in SFX-TEB in JJA might be explained by the model's lack of coupling with the atmosphere but it is difficult to disentangle this lack of coupling from other possible sources of error (e.g. misrepresentation of surface properties, observational uncertainty). There is certainly plenty of room for improvement in the modeling of urban physical processes, but this is beyond the scope of this study.

**- I expect that UHI is dominant in nighttime temperature because of thermal properties of buildings but Fig 3 and 5 show that SFX-TEB does not give substantial improvement in nighttime. Could you provide any idea and implications on this?**

R: There is not a substantial improvement in nighttime because the offline approach does not include land-atmosphere feedbacks. We mentioned this fact in the conclusions section (lines 482-484).

**7. Figure 7: It is interesting that the SFX-TEB improves the UHI near sunrise and sunset substantially. Could you discuss this finding?**

R: SURFEX overestimates in daytime and underestimates in nighttime (due to lack of land-atmosphere feedbacks). The sunset/sunrise improvement is likely due to SFX-TEB getting the timing of the cooling/warming correctly and the feedbacks being less impactful at those times of the day.

**8. Figure 2 and 8: Can you add urban boundary described in Figure into these two figures?**

R: We added the inner circle from Figure 1 that represents the urban boundary to Figures 2 and 9 (old Figure 8).